# THREE DIMENSIONAL RECONSTRUCTION OF BOTANICAL TREES WITH SIMULATABLE GEOMETRY

## ABSTRACT

We tackle the challenging problem of creating full and accurate three dimensional reconstructions of botanical trees with the topological and geometric accuracy required for subsequent physical simulation, e.g. in response to wind forces. Although certain aspects of our approach would benefit from various improvements, our results exceed the state of the art especially in geometric and topological complexity and accuracy. Starting with two dimensional RGB image data acquired from cameras attached to drones, we create point clouds, textured triangle meshes, and a simulatable and skinned cylindrical articulated rigid body model. We discuss the pros and cons of each step of our pipeline, and in order to stimulate future research we make the raw and processed data from every step of the pipeline as well as the final geometric reconstructions publicly available.

## 1 INTRODUCTION

Human-inhabited outdoor environments typically contain ground surfaces such as grass and roads, transportation vehicles such as cars and bikes, buildings and structures, and humans themselves, but are also typically intentionally populated by a large number of trees and shrubbery; most of the motion in such environments comes from humans, their vehicles, and wind-driven plants/trees. Tree reconstruction and simulation are obviously useful for AR/VR, architectural design and modeling, film special effects, etc. For example, when filming actors running through trees, one would like to create virtual versions of those trees with which a chasing dinosaur could interact. Other uses include studying roots and plants for agriculture (Zheng et al., 2011; Estrada et al., 2015; Fuentes et al., 2017) or assessing the health of trees especially in remote locations (similar in spirit to Zuffi et al. (2018)). 2.5D data, i.e. 2D images with some depth information, is typically sufficient for robotic navigation, etc.; however, there are many problems that require true 3D scene understanding to the extent one could 3D print objects and have accurate geodesics. Whereas navigating around objects might readily generalize into categories or strategies such as 'move left,' 'move right,' 'step up,' 'go under,' etc., the 3D object understanding required for picking up a cup, knocking down a building, moving a stack of bricks or a pile of dirt, or simulating a tree moving in the wind requires significantly higher fidelity. As opposed to random trial and error, humans often use mental simulations to better complete a task, e.g. consider stacking a card tower, avoiding a falling object, or hitting a baseball (visualization is quite important in sports); thus, physical simulation can play an important role in end-to-end tasks, e.g. see Kloss et al. (2017); Peng et al. (2017); Jiang & Liu (2018) for examples of combining simulation and learning.

Accurate 3D shape reconstruction is still quite challenging. Recently, Malik argued[1] that one should not apply general purpose reconstruction algorithms to say a car and a tree and expect both reconstructions to be of high quality. Rather, he said that one should use domain-specific knowledge as he has done for example in Kanazawa et al. (2018). Another example of this specialization strategy is to rely on the prior that many indoor surfaces are planar in order to reconstruct office spaces (Huang et al., 2017) or entire buildings (Armeni et al., 2016; 2017). Along the same lines, Zuffi et al. (2018) uses a base animal shape as a prior for their reconstructions of wild animals. Thus, we similarly take a specialized approach using a generalized cylinder prior for both large and medium scale features.

In Section 3, we discuss our constraints on data collection as well as the logistics behind the choices we made for the hardware (cameras and drones) and software (structure from motion, multi-view

---

[1]Jitendra Malik, Stanford cs231n guest lecture, 29 May 2018

stereo, inverse rendering, etc.) used to obtain our raw and processed data. Section 4 discusses our use of machine learning, and Section 5 presents a number of experimental results. In Appendices A, B, and C we describe how we create geometry from the data with enough efficacy for physical simulation.

## 2 Previous Work

**Tree Modeling and Reconstruction:** Researchers in computer graphics have been interested in modeling trees and plants for decades (Lindenmayer, 1968; Bloomenthal, 1985; Weber & Penn, 1995; Prusinkiewicz et al., 1997; Stava et al., 2014). SpeedTree[2] is probably the most popular software utilized, and their group has begun to consider the incorporation of data-driven methods. Amongst the data-driven approaches, Tan et al. (2007) is most similar to ours combining point cloud and image segmentation data to build coarse-scale details of a tree; however, they generate fine-scale details procedurally using a self-similarity assumption and image-space growth constraints, whereas we aim to capture more accurate finer structures from the image data. Other data-driven approaches include Livny et al. (2010) which automatically estimates skeletal structure of trees from point cloud data, Xie et al. (2015) which builds tree models by assembling pieces from a database of scanned tree parts, etc.

Many of these specialized, data-driven approaches for trees are built upon more general techniques such as the traditional combination of structure from motion (see e.g. Wu (2013)) and multi-view stereo (see e.g. Furukawa & Ponce (2010)). In the past, researchers studying 3D reconstruction have engineered general approaches to reconstruct fine details of small objects captured by sensors in highly controlled environments (Seitz et al., 2006). At the other end of the spectrum, researchers have developed approaches for reconstructing building- or even city-scale objects using large amounts of image data available online (Agarwal et al., 2009). Our goal is to obtain a 3D model of a tree with elements from both of these approaches: the scale of a large structure with the fine details of its many branches and twigs. However, unlike in general reconstruction approaches, we cannot simply collect images online or capture data using a high-end camera.

To address similar challenges in specialized cases, researchers take advantage of domain-specific prior knowledge. Zhou et al. (2008) uses a generalized cylinder prior (similar to us) for reconstructing tubular structures observed during medical procedures and illustrates that this approach performs better than simple structure from motion. The process of creating a mesh that faithfully reflects topology and subsequently refining its geometry is similar in spirit to Xu et al. (2018), which poses a human model first via its skeleton and then by applying fine-scale deformations.

**Learning and Networks:** So far, our use of networks is limited to segmentation tasks, where we rely on segmentation masks for semi-automated tree branch labeling. Due to difficulties in getting sharp details from convolutional networks, the study of network-based segmentation of thin structures is still an active field in itself; there has been recent work on designing specialized multiscale architectures (Ronneberger et al., 2015; Lin et al., 2017; Qu et al., 2018) and also on incorporating perceptual losses (Johnson et al., 2016) during network training (Mosinska et al., 2018).

## 3 Raw and Processed Data

As a case study, we select a California oak (*quercus agrifolia*) as our subject for tree reconstruction and simulation (see Figure 1). The mere size of this tree imposes a number of restrictions on our data capture: one has to deal with an outdoor, unconstrained environment, wind and branch motion will be an issue, it will be quite difficult to observe higher up portions of the tree especially at close proximities, there will be an immense number of occluded regions because of the large number of branches that one cannot see from any feasible viewpoint, etc.

In an outdoor setting, commodity structured light sensors that use infrared light (e.g. the Kinect) fail to produce reliable depth maps as their projected pattern is washed out by sunlight; thus, we opted to use standard RGB cameras. Because we want good coverage of the tree, we cannot simply capture images from the ground; instead, we mounted our cameras on a quadcopter drone that was piloted around the tree. The decision to use a drone introduces additional constraints: the cameras must be

---

[2]https://speedtree.com

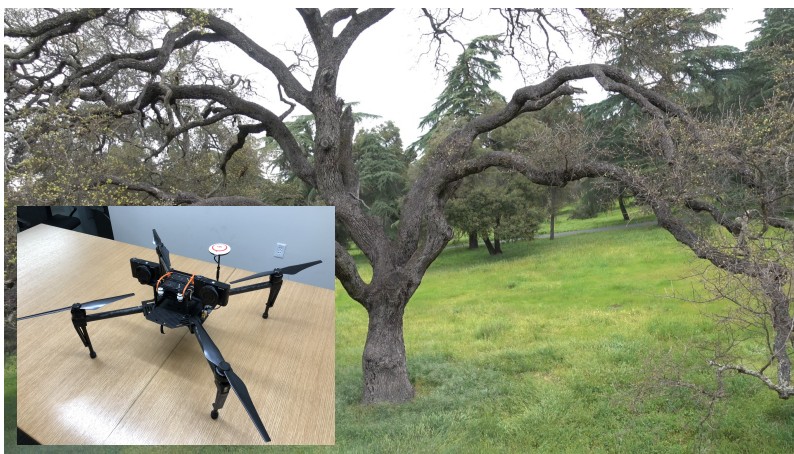

Figure 1: We target a California oak for reconstruction and simulation. (Inset) The drone and camera setup used to collect video data of the tree.

lightweight, the camera locations cannot be known *a priori*, the drone creates its own air currents which can affect the tree's motion, etc. Balancing the weight constraint with the benefits of using cameras with a global shutter and minimal distortion, we mounted a pair of Sony rx100 v cameras to a DJI Matrice 100 drone. We calibrated the stereo offset between the cameras before flight, and during flight each camera records a video with 4K resolution at 30 fps.

Data captured in this manner is subject to a number of limitations. Compression artifacts in the recorded videos may make features harder to track than when captured in a RAW format. Because the drone must keep a safe distance from the tree, complete 360° coverage of a given branch is often infeasible. This lack of coverage is compounded by occlusions caused by other branches and leaves (in seasons when the latter are present). Furthermore, the fact that the tree may be swaying slightly in the wind even on a calm day violates the rigidity assumption upon which many multi-view reconstruction algorithms rely. Since we know from the data collection phase that our data coverage will be incomplete, we will need to rely on procedural generation, inpainting, "hallucinating" structure, etc. in order to complete the model.

After capturing the raw data, we augment it to begin to estimate the 3D structure of the environment. We subsample the videos at a sparse 1 or 2 fps and use the Agisoft PhotoScan tool[3] to run structure from motion and multi-view stereo on those images, yielding a set of estimated camera frames and a dense point cloud. We align cameras and point clouds from separate structure from motion problems by performing a rigid fit on a sparse set of control points. This is a standard workflow also supported by open-source tools (Wu, 2011; Schönberger & Frahm, 2016; Moulon et al., 2016). Some cameras may be poorly aligned (or in some cases, so severely incorrect that they require manual correction). Once the cameras are relatively close, one can utilize an inverse rendering approach like that of Loper & Black (2014) adjusting the misaligned cameras' parameters relative to the point cloud. In the case of more severely misaligned cameras, one may select correspondences between 3D points and points in the misaligned image and then find the camera's extrinsics by solving a perspective-*n*-point problem (Fischler & Bolles, 1981).

In the supplemental appendices, we describe our approach to constructing large scale geometry using this processed data. Recovering "medium" scale structures that are not captured in the point cloud, however, is a problem that lends itself well to a learning-based treatment.

## 4    ANNOTATION AND LEARNING

Annotating images is a challenging task for human labelers and automated methods alike. Branches and twigs heavily occlude one another, connectivity can be difficult to infer, and the path of even a relatively large branch can often not be traced visually from a single view. Thus it is desirable to augment the image data during annotation to aid human labelers.

---

[3]Agisoft PhotoScan, http://www.agisoft.com/

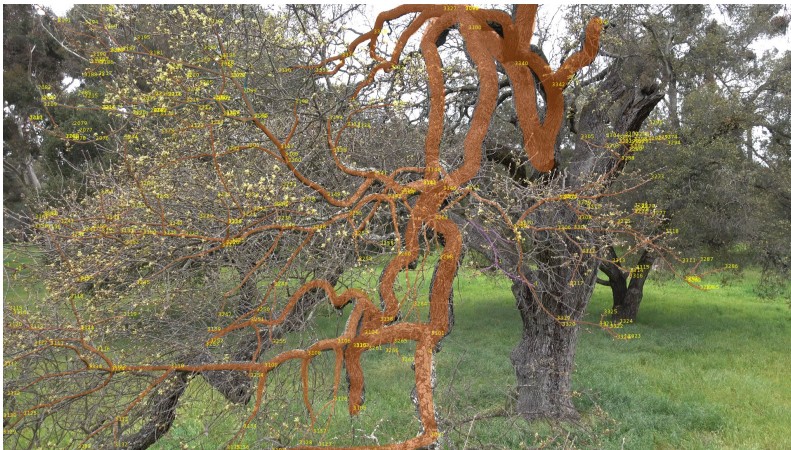

Figure 2: Human labelers use our annotation tool to draw curves with positions, thicknesses, connectivities, and unique identifiers on images of the tree.

One method for aiding the labeler is to automatically extract a "flow field" of vectors tracking the anisotropy of the branches in image space (see Figure 6). The flow field is overlaid on the image in the annotation tool, and the labeler may select endpoints to be automatically connected using the projection-advection scheme discussed in Section 5.3. Section 5.3 also discusses how we generate the flow field itself, after first creating a segmentation mask. Note that segmentation (i.e. discerning *tree* or *not tree* for each pixel in the image) is a simpler problem than annotation (i.e. discerning medial axes, topology, and thickness in image space).

Obtaining segmentation masks is straightforward under certain conditions, e.g. in areas where branches and twigs are clearly silhouetted against the grass or sky, but segmentation can be difficult in visually dense regions of an image. Thus, we explore deep learning-based approaches for performing semantic segmentation on images from our dataset. In particular, we use U-Net (Ronneberger et al., 2015), a state-of-the-art fully convolutional architecture for segmentation; the strength of this model lies in its many residual connections, which give the model the capacity to retain sharp edges despite its hourglass structure. See Section 5.2 for further discussion.

## 5 EXPERIMENTS

Since the approach to large scale structure discussed in Appendix A works sufficiently well, we focus here on medium scale branches.

### 5.1 IMAGE ANNOTATION

We present a human labeler with an interface for drawing piecewise linear curves on an overlay of a tree image. User annotations consist of vertices with 2D positions in image space, per-vertex branch thicknesses, and edges connecting the vertices. Degree-1 vertices are curve endpoints, degree-2 vertices lie on the interior of a curve, and degree-3 vertices exist where curves connect. A subset of the annotated vertices are additionally given unique identifiers that are used to match common points between images; these will be referred to as "keypoints" and are typically chosen as bifurcation points or points on the tree that are easy to identify in multiple images. See Figure 2.

We take advantage of our estimated 3D knowledge of the tree's environment in order to aid human labelers and move towards automatic labeling. After some annotations have been created, their corresponding 3D structures are generated and projected back into each image, providing rough visual cues for annotating additional images. Additionally, since we capture stereo information, we augment our labeling interface to be aware of stereo pairs: users annotate one image, copy those annotations to the stereo image, and translate the curve endpoints along their corresponding epipolar lines to the correct location in the stereo image. This curve translation constrained to epipolar lines (with additional unconstrained translation if necessary to account for error) is much less time consuming than labeling the stereo image from scratch.

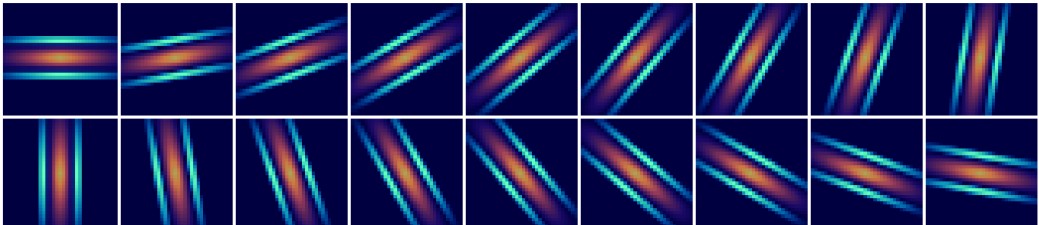

Figure 3: A set of anisotropic kernels is used to obtrain directional activations in segmentation masks for both perceptual loss and flow field generation.

Human labelers often identify matching branches and twigs across images by performing human optical flow, toggling between adjacent frames of the source video and using the parallax effect to determine branch connectivity. This practice is an obvious candidate for automation, e.g. by annotating an initial frame then automatically carrying the annotated curves through subsequent frames via optical flow. Unfortunately, the features of interest are often extremely small and thin and the image data contains compression artifacts, making automatic optical flow approaches quite difficult. However, it is our hope that in future work the same tools that aid human labelers can be applied to automatic approaches making them more effective for image annotation.

## 5.2 DEEP LEARNING

In order to generate flow fields for assisting the human labeler as discussed in Section 4, we first obtain semantic segmentations of *tree* and *not tree* using a deep learning approach. To train a network for semantic segmentation, we generate a training dataset by rasterizing the image annotations as binary segmentation masks of the labeled branches. From these 4K masks, we then generate a dataset of $512 \times 512$ crops containing more than 4000 images. The crop centers are guaranteed to be at least 50 pixels away from one another, and each crop is guaranteed to correspond to a segmentation mask containing both binary values. The segmentation problem on the raw 4K images must work on image patches with distinctly different characteristics: the more straightforward case of branches silhouetted against the grass, and the more complex case of highly dense branch regions. Therefore, we split the image patches into two sets via $k$-means clustering, and train two different models to segment the two different cases. For the same number of training epochs, our two-model approach yields qualitatively better results than the single-model approach.

Instead of directly using the standard binary cross entropy loss, the sparseness and incompleteness of our data led us to use a weighted variant, in order to penalize false negatives more than false positives. As a further step to induce smoothness and sparsity in our results, we introduce a second order regularizer through the L2 difference of the output and ground truth masks' gradients. We also experiment with an auxiliary loss similar to the VGG perceptual loss described in Mosinska et al. (2018), but instead of using arbitrary feature layers of a pretrained network, we look at the L1 difference of hand-crafted multiscale directional activation maps. These activation maps are produced by convolving the segmentation mask with a series of Gabor filter-esque (Jain & Farrokhnia, 1991) feature kernels $\{k(\theta, r, \sigma) : \mathbb{R}^2 \to [0, \dots, N]^2\}$, where each kernel is scale-aware and piecewise sinusoidal (see Figure 3). A given kernel $k(\theta, r, \sigma)$ detects branches that are at an angle $\theta$ and have thicknesses within the interval $[r, \sigma r]$. For our experiments, we generate 18 kernels spaced 10 degrees apart and use $N = 35$, $r = 4$, and $\sigma = 1.8$.

Figure 4 illustrates two annotated images used in training and the corresponding learned semantic segmentations. Note that areas of the semantic segmentation that are not part of the labeled annotation may correspond to true branches or may be erroneous; for the time being a human must still choose which pieces of the semantic segmentation to use in adding further annotations.

## 5.3 LEARNING-ASSISTED ANNOTATION

To generate a flow field, we create directional activation maps as in Section 5.2 again using the kernels from Figure 3, then perform a clustering step on the resulting per-pixel histograms of gradients (Dalal & Triggs, 2005) to obtain flow vectors. Each pixel labeled as *tree* with sufficient confidence is assigned one or more principal directions; pixels with more than one direction are

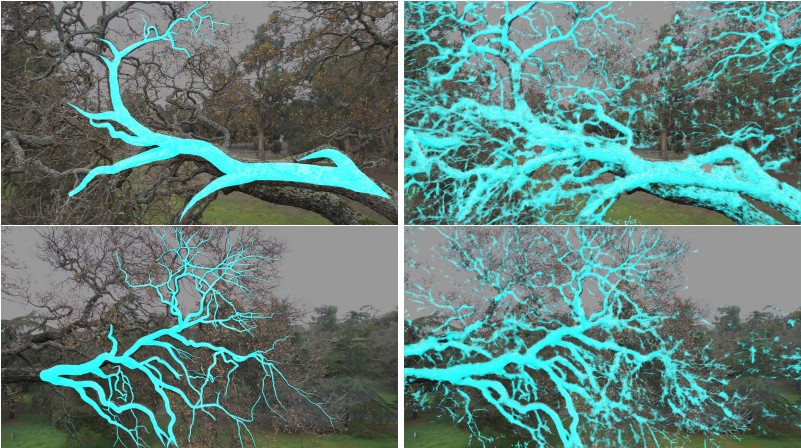

Figure 4: (Left) Image masks generated from image annotations and used as training data. (Right) Outputs of the segmentation network.

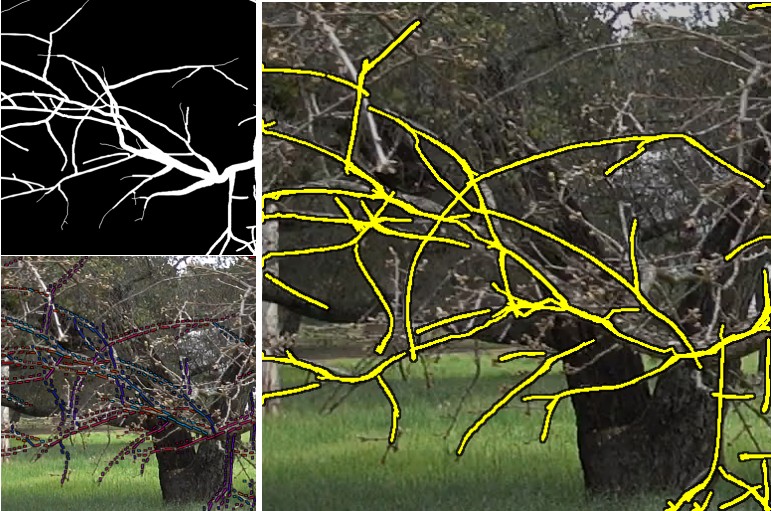

Figure 5: (Top left) A ground truth mask of the tree taken by flattening the image annotation data into a simple binary mask. (Bottom left) A visualization of flow directions estimated by applying directional filters to the ground truth mask. (Right) Medial axes of the tree branches estimated from the flow field.

potentially branching points. We find the principal directions by detecting clusters in each pixel's activation weights; for each cluster, we take the sum of all relevant directional slopes weighted by their corresponding activation values.

Having generated a flow field of sparse image space vectors, we trace approximate medial axes through the image via an alternating projection-advection scheme. From a given point on a branch, we estimate the thickness of the branch by examining the surrounding flow field and project the point to the estimated center of the branch. We then advect the point through the flow field and repeat this process. In areas with multiple directional activations (e.g. at branch crossings or bifurcations), our advection scheme prefers the direction that deviates least from the previous direction. More details about this scheme may be found in the supplemental material. By applying this strategy to flow fields generated from ground truth image segmentations, we are able to recover visually plausible medial axes (see Figure 5). However, medial axes automatically extracted from images without ground truth labels are error prone. Thus, we overlay the flow field on the annotation interface and rely on the human labeler. The labeler may select curve endpoints in areas where the flow field is visually plausible, and these endpoints are used to guide the medial axis generation. See Figure 6 for an example flow field generated from the learned segmentation mask and the supplemental material for a demonstration of semi-automated medial axis generation.

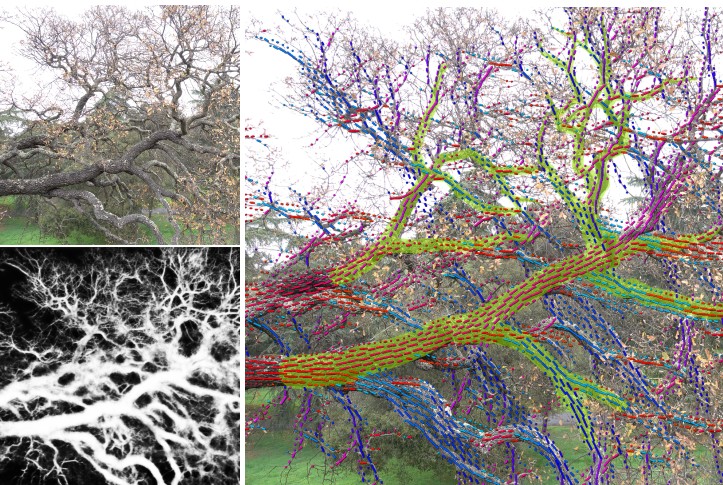

Figure 6: The trained network infers a segmentation mask (bottom left) from an input image (top left). We then estimate a flow field (right) by applying anisotropic filters to the segmentation mask. A labeler can specify endpoints between which a medial axis and thickness values are automatically estimated (right, in green).

### 5.4 RECOVERING MEDIUM SCALE BRANCHES

Given a set of image annotations and camera extrinsics obtained via structure from motion and stereo calibration, we first construct piecewise linear branches in 3D. We triangulate keypoints that have been labeled in multiple images, obtaining 3D positions by solving for the point that minimizes the sum of squared distances to the rays originating at each camera's optical center and passing through the camera's annotated keypoint. We then transfer the topology of the annotations to the 3D model by connecting each pair of 3D keypoints with a line segment if a curve exists between the corresponding keypoint pair in any image annotation.

Next, we subdivide and perturb the linear segments connecting the 3D keypoints to match the curvature of the annotated data. Each segment between two keypoints is subdivided by introducing additional vertices evenly spaced along the length of the segment. For each newly introduced vertex, we consider the point that is the same fractional length along the image-space curve between the corresponding annotated keypoints in each image for which such a curve exists. We trace rays through these intra-curve points to triangulate the position of each new vertex in the same way that we triangulated the original keypoints.

Finally, we estimate the thickness of each 3D vertex beginning with the 3D keypoints. We estimate the world space thickness of each keypoint by considering the corresponding thickness in all annotated camera frames. For each camera in which the keypoint is labeled, we estimate world space thickness using similar triangles, then average these estimates to get the final thickness value. We then set the branch thickness of each of the vertices obtained through subdivision simply by interpolating between the thicknesses of the keypoints at either end of the 3D curve. Using this strategy, we recover a set of 3D positions with local cross-sectional thicknesses connected by edges, which is equivalent to the generalized cylinder representation employed in Appendix A.

The human users of our annotation tools encounter the traditional trade-off of stereo vision: it is easy to identify common features in images with a small baseline, but these features triangulate poorly exhibiting potentially extreme variance in the look-at directions of the corresponding cameras. Conversely, cameras whose look-at directions are close to orthogonal yield more stable triangulations, but common features between such images are more difficult to identify. One heuristic approach is to label each keypoint three times: twice in similar images and once from a more diverse viewpoint. However, it may be the case that some branches are only labeled in two images with a small baseline (e.g. a stereo pair). In this case, we propose a clamping strategy based on the topological prior of the tree. Designating a "root" vertex of a subtree for such annotations, we triangulate the annotated keypoints as usual obtaining noisy positions in the look-at directions of the stereo cameras. We then march from the root vertex to the leaf vertices. For each vertex $p$ with location $p_x$, we consider

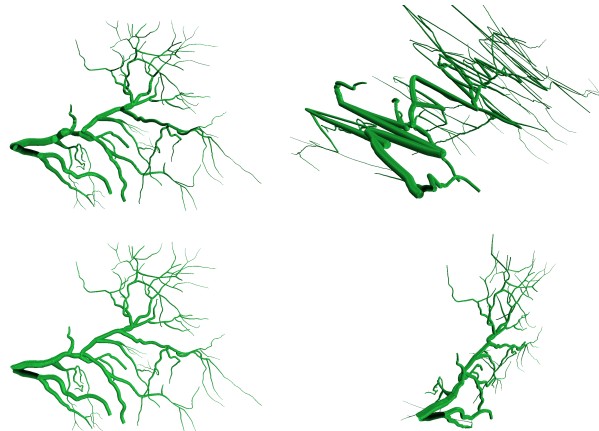

Figure 7: Branches labeled from a stereo pair of cameras are visually plausible from the perspective of those cameras (top left), but they can exhibit severe error when viewed from a different angle (top right). By clamping these branch positions, one can achieve a virtually identical projection to the original cameras (bottom left) while maintaining a nondegenerate albeit "flattened" appearance from a different angle (bottom right).

each outboard child vertex $c$ with location $c_x$. For each camera in which the point $c$ is labeled, we consider the intersection of the ray from the location of $c$'s annotation to $c_x$ with the plane parallel to the image plane that contains $p_x$; let $c'_x$ be the intersection point. We then clamp the location of $c$ between $c'_x$ and the original location $c_x$ based on a user-specified fraction. This process is repeated for each camera in which $c$ is annotated, and we obtain the final location for $c$ by averaging the clamped location from each camera. See Figure 7.

## 6 CONCLUSION AND FUTURE WORK

We presented an end-to-end pipeline for reconstructing a 3D model of a botanical tree from RGB image data. Our reconstructed model may be readily simulated to create motion in response to external forces, e.g. to model the tree blowing in the wind (see Figure 8). We use generalized cylinders to initialize an articulated rigid body system, noting that one could subdivide these primitives as desired for additional bending degrees of freedom, or decrease their resolution for faster performance on mobile devices. The simulated bodies drive the motion of the textured triangulated surfaces and/or the point cloud data as desired.

Although we presented one set of strategies to go all the way from the raw data to a simulatable mesh, it is our hope that various researchers will choose to apply their expertise to and improve upon various stages of this pipeline, yielding progressively better results. In particular, the rich topological information in the annotations has great potential for additional deep learning applications, particularly for topology extraction (Ventura et al., 2018; Máttyus et al., 2017; Xue et al., 2018) and 2D to 3D topology generation (Estrada et al., 2015).

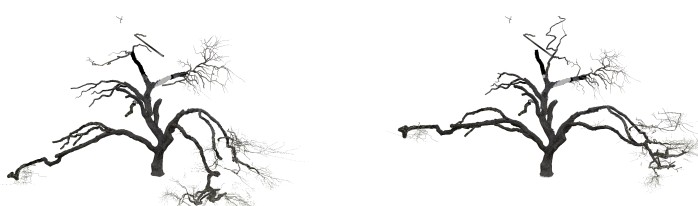

Figure 8: The tree model is deformed from its rest pose (left) to an exaggerated pose (right) via simulation.

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
