# OpenReview forum: "Three Dimensional Reconstruction of Botanical Trees with Simulatable Geometry"
_ICLR.cc/2021/Conference — Reject_

### Official Review · AnonReviewer4 · 2020-10-25
**Paper tackles interesting and challenging problem and proposes a sound solution, very limited evaluation, no ablation study, importance of learning unclear**

**Rating:** 4
**Confidence:** 4

**Review:**

The paper considers an interesting and challenging problem, namely reconstructing 3D structures with very fine details from image data captured "in-the-wild", i.e., taken in uncontrolled environments. More concretely, the paper considers the problem of obtaining a detailed 3D reconstruction of trees that results in a 3D model that can be animated.

The paper proposes a sound pipeline suitable for the task at hand: first, images of the tree are captured from cameras mounted on a drone to allow capturing the tree from a wide range of angles. Structure-from-Motion is then used to recover the larger structures in the tree from these images. Next, finer structured are recovered. An annotator first annotates branches in some of the images. The known 3D structure is thereby used to aid with the annotation task by using the known epipolar geometry. These labels are then used to train two U-Net architectures (one to classify branches in front of a grassy ground and the other to handle denser regions with multiple branches). Applying these networks on the images then provides initial segmentations that are used to aid the annotator. In particular, a flow field is estimated to propagate information about branches and their thickness. Given the annotation, the corresponding 3D structures are then estimated using triangulation, using a form a smoothness prior on the branch geometry.

The proposed approach is sound and seems suited for the task at hand. However, I am not fully convinced that one would want to apply it in practice for multiple reasons:
1) As stated in the introduction, using general purpose 3D reconstruction approaches can lead to sub-optimal results when constructing harder structures such as trees. Thus, the paper starts from a general purpose method (Structure-from-Motion) and adds processing steps designed to recover detailed tree branches. However, it is unclear how it compares against non-general purpose 3D reconstruction methods. For example, [Häne et al., Joint 3D scene reconstruction and class segmentation, CVPR 2013] and [Häne et al., Class Specific 3D Object Shape Priors Using Surface Normals, CVPR 2014] show that using class-specific priors can significantly improve reconstruction quality.
There is work on reconstructing fine details based on connectivity information, e.g., [Stühmer et al., Tree Shape Priors with Connectivity Constraints using Convex Relaxation on General Graphs, ICCV 2013] and [Oswald et al., Generalized Connectivity Constraints for Spatio-temporal 3D Reconstruction, ECCV 2014]. These approaches seem to be very suitable for the task at hand. Unfortunately, these methods are not even discussed in the paper.
[Yücher et al., Efficient 3D Object Segmentation from Densely Sampled Light Fields with Applications to 3D Reconstruction, ACM TOG 2016] and [Yücher et al., Depth from Gradients in Dense Light Fields for Object Reconstruction, 3DV 2016] show impressive results for reconstructing fine details from light-fields (e.g., obtained via densely sampled videos), including recovering small branches of vegetation. These methods also seem to be applicable to the problem at hand. Unlike the proposed approach, they do not seem to require manual annotation.
None of these methods, which seem suitable alternatives for the proposed approach, are discussed in the paper or compared against.
2) The importance of the individual steps of the proposed pipeline is unclear to me:
  a) How much does training the networks for segmentation help with the workflow, i.e., by using a network to aid the annotator, how much more human time is saved? Ideally, the paper would provide run-times for the individual stages to show the importance of this task.
  b) How important is it to train two segmentation networks instead of a single one? Unfortunately, the paper does not provide an ablation study here.
  c) How does the proposed approach of using learning compare to advanced annotation tools such as [Acuna et al., Efficient Interactive Annotation of Segmentation Datasets with Polygon-RNN++, CVPR 2018]?
3) It is unclear to me how general the proposed approach is. Can it be readily be applied to other types of trees? Or would it require significant changes? Unfortunately, the paper only provides results for a single tree of a single species.

In my opinion, the presentation of the proposed method can be significantly improved:
1) Given the title and abstract, I would have expected the paper to discuss how simulatable geometry is estimated. In addition, the need for simulatable geometry features prominently in the introduction. However, this part is never really explained in the paper and the paper mostly refers to the supplementary material.
2) Given that the paper is about reconstructing trees, I would have expected that it shows some qualitative or quantitative results for the reconstructed trees. Qualitative results are shown in the supplementary material, but not in the paper.
3) In general, the paper is rather light on details. I had the impression that the supplementary material (both the pdf and the video) provide a better explanation of the proposed method than the main paper (at least watching the video helped me to better understand how the proposed approach is structured and how the individual stages depend on each other). It would be good to move some of these details and descriptions from the supplementary material to the paper.
4) I find the name of the "Experiments" section a bit misleading as it mostly discusses parts of the method and does not show much experimental results.
5) It is unclear to me why one would not simply use a hardware-synchronized stereo camera instead of two individual cameras. Hardware synchronization would remove problems such as slight motion of the tree between the two images. One could even use a multi-camera system such as the one from [Gohl et al., Omnidirectional visual obstacle detection using embedded FPGA, IROS 2015] to be able to see a larger part of the tree.

Overall, I think the paper considers an interesting problem and proposes a sensible solution. However, the proposed method currently stands rather isolated, both compared to previous work applicable in this setting and in terms of exploring alternatives for some of its components / showing the importance of its components. Together with problems in the presentation, I feel that the paper in its current form is not yet ready for publication and would require a significant revision (improved presentation, comparisons with existing work, more detailed experimental evaluation).

### After rebuttal phase ###
Since the authors did not provide a revised version of the paper or addressed my comments in detail, I do not see a reason to change my initial recommendation. I thus recommend to reject the paper.

---

> ### Author Response · Authors · 2020-11-25
> **Response to Reviewer 4**
>
> Thank you for your comments.
>
> We would be happy to examine and add discussion of the mentioned related works. It is certainly possible that some of these might be used to improve upon various stages of the overall pipeline! Our goal is to perform in-depth data collection for a single tree and to make that data available, but we certainly do not believe this submission is the final word on this reconstruction problem.
>
> The observation that some aspects of the paper require referring to supplemental material is also a very fair point--and indeed a consequence of us reworking the paper to target ICLR after a previous submission. We would be happy to edit the exposition to address the points you mention.

---

### Official Review · AnonReviewer3 · 2020-10-25
**3D visual reconstruction of tree/branches in an interesting and important task.   This paper described a complete pipeline for accurately recovering geometrically and topologically accurate tree structures by learning based annotation.**

**Rating:** 4
**Confidence:** 3

**Review:**

This paper tackles the problem of geometrical and topological 3D reconstruction of a (botanical) tree using a drone-mounted stereo vision system and deep learning-based/aided tree branch image annotation procedures.   This is an interesting computer vision 3D reconstruction task, which has important practical applications (e.g in AR/VR or for plant phonemics study), however has not been extensively researched in the past.   Part of the reasons are due to some unique challenges that the problem of  tree reconstruction is facing, in particular, how to accurately recover complex visual occlusions caused by dense tree branches and leaves, and how to ensure the reconstructed topology is accurate.

The paper presents a working system, a pipeline, that the authors have developed.  Instead of describing the procedures for recovering  3D geometry reconstruction of the tree, the authors focused their attention in this paper on the recovery of (medium scale) tree branches structure , such as  branch semantic segmentation, tree truck and brach width and direction estimation, as well as the topology reasoning.     They approach such medium-scale reasoning task through a combined used of supervised learning, and semi-automated manual labelling.    Specifically, a binary semantic-segmentation task to separate object from background (i.e.  tree from non-tree) is conducted via a U-Net type neural networks,  followed by a tree branch orientation) estimation  by applying a hand-crafted anisotropic orientation filter banks.   These orientation filters banks have a  Gabor-like steerable filter shape.  Outputs (or activations) of such orientation filters, i.e. the estimated  flow field,  are used to infer the dominating directions of the tree branch in question.  Finally, other geometric parameters (such as branch width, median axis, and cluster membership) are estimated to conclude the medium-scale structure recovery task.

The method is tested on a limited number of real images, and impressive results are obtained.

Pluses:
- interest and important problem, not been extensively investigated in the past.
- impressive reconstruction results, as demonstrated in the companioning video.
- standard reconstruction pipeline, easy to follow.

Minus:
 -  Ad hoc, heavy-handed labelling  pipeline;  strong in engineering, but weak in original scientific contribution.  The paper feels more like an experimental report suitable for ICRA/IROS, and less of a scientific paper in the field of Representation Learning or ICLR.
- limited theoretical or methodological contribution.  Reading the paper won't learn much novel insights in solving similar tasks.
-  Insufficient experiments. It seems only one instance of test is reported/presented in the paper and in the supp. material.
-  The entire labelling process is not an end-to-end solution: rather it requires constant human intervention.

---

> ### Author Response · Authors · 2020-11-25
> **Response to Reviewer 3**
>
> We thank the reviewer for their comments and feedback.
>
> It is certainly true that this pipeline is heavily tied to human interaction. Indeed we believe that our state-of-the-art is not yet at a point where a quality end-to-end solution is available. However, we hope that by doing the labor-intensive work, even if only for a single instance of a real tree, and supplying the resulting data to the community, we may provide a complete pipeline within which other researchers can work and which they can build upon.

---

### Official Review · AnonReviewer2 · 2020-10-28
**Digital Dendritic Decals**

**Rating:** 6
**Confidence:** 3

**Review:**

The authors describe a processing pipeline for producing 3D models of trees from stereo visual data, which is later extended with a DNN for segmentation.

The paper is more on the practical, or application side, although very interesting especially for tackling a very complex task, which was bound to present challenges since segmenting thing and long structures is often hard, as the authors mention.

This kind of paper is often challenging to review, because it is clear that a lot of good work has been done, and the overarching goal of the project seems interesting. It is hard to assess the paper, though, if apart from reporting on the methods the results are only final models obtained. It might suffice, although the authors should be strongly encouraged to find some kind of experiment that could have been done to validate the results more. For example, there could be objective assessments of the models to see if they fit some kind of physical prediction about the trees shape and laws of physics. Or perhaps the labeling procedure could have been investigated, and the segmentation shown to be accurate and to accelerate the labeling. Even a small result, as long as there is some kind of objective analysis that can be fit in a hypothesis-testing framework, can go a long way. How do I know these models produced are any good, because I looked at the paper and found them nice? It is very important when writing a scientific paper to make sure some kind of objective analysis can be offered.

It is also important to make the proposal very clear from the beginning. The title, abstract and first section of the paper focus too much on the long term intentions of the authors, instead of objectively and concisely describing exactly what pertains to the paper. The authors should try to fit in the abstract a short description of the processing pipeline that explains that stereo visual data was used to label the tree images, then the labels used to train a U-net with Gabor filters, and this used again to speed up labeling, for example. Whatever is the gist of it, this should be soon in the abstract and everywhere in the paper. Avoid making a mystery of what is the work done, on the contrary, repeat many times what has been done, completely, in different levels of detail.

The recommendation is to accept the paper mostly by virtue of the authors approaching an interesting problem and having produced some results. It's only marginal, though, because the papers still feels a little incomplete an perhaps lacks novelty.

Some specific remarks:

What makes the model "simulable"? How do we know it was achieved? Or was it a constraint from the beginning?

Why call it "Gabor filter-esque", how is it not exactly a Gabor filter bank? And a quick note about the filters, have you considered also using quadrature filters, with odd symmetry? The even-symmetry filters used may be more adequate for detecting ridges than edges, it might be nice to have both, as proposed in many papers in edge-detection literature.

Use of classic function bases and filter banks with deep learning is a very interesting, and maybe one of the most interesting aspects of this paper. Perhaps cite other examples of research where this has been done, one example that comes to mind is https://arxiv.org/pdf/1801.10130.pdf

It seems the labels were only 2D, always, although 3D information was used to greatly help in the process. Why not actually label as 3D?

The neural network was only employed in 2D, even though much 3D data is available. Could the model have benefited from taking stereo pairs as input, perhaps?

One very interesting aspect of 3D reconstruction is that sometimes the geometry of objects in 2D is more complicated than in 3D, because we are looking at data distorted by perspective, etc. It would be very interesting to see how this work could benefit from enforcing priors in 3D. Right now the approach still seems to be mostly treating the problem as 2D. What potential do the authors see in leveraging more 3D information?

The model is said to be based on cylinders, although if the thickness is changing along the segments, is it not more accurately cones?

One very important topological constraint of the peculiar objects studied is that they appear to create graphs that contain no cycles (!) might be good to mention this interesting fact somewhere in the paper.

---

> ### Author Response · Authors · 2020-11-25
> **Response to Reviewer 2**
>
> Thank you for your feedback!
>
> Responding to specific remarks:
> We consider our cylinder-based model "simulatable" because we can directly provide the reconstructed cylinders and their parent-child relationships to any number of physically-based solvers or animation tools. Contrast this with e.g. a raw point cloud, which is not simulation-ready because it carries no knowledge of the tree's topology or the connections between points. Even processing such a point cloud without domain specific-knowledge of the tree's structure (e.g. using the poisson surface reconstruction commonly available in software packages) can result in disconnected components that would thus not behave correctly in a simulation.
>
> Our filters are Gabor-"esque" and not exactly Gabor filters because instead of multiplying the sinusoidal term by a Gaussian kernel, which does not have finite support, we multiply by a very simple piecewise linear function in order to have zero activation if an image patch contains a line that is thicker than our maximum-detectable thickness (appendix G in the supplemental material). We use these filters to detect "elongated blobs" rather than the edges on either side of a branch, then to subsequently trace a centerline using a projection-advection scheme.
> After reading up on quadrature filters based on this review, we believe that our filter in appendix G may actually be a type of quadrature filter. We would be happy to learn more about this and modify the exposition as appropriate, and to include discussion of the mentioned paper -- we could potentially benefit from the techniques these authors use to handle rotational invariance.
>
> The suggestion regarding the use of stereo pairs in network training is interesting -- admittedly we use stereo-assisted annotation and network-assisted annotation as distinct techniques for semi-automated annotation. Thus far, we have used the U-net network architecture to learn segmentations on images. However, with respect to 3D data, we believe it would be very interesting to use graph convolutional networks to infer the topological structure of the 3D tree.
>
> Cylinder vs cones -- yes, we use the term "generalized cylinder" to mean a cylinder with two different radii; we might instead say "conical frustum" to be more precise.
>
> Graphs without cycles -- yes! Trees in the botanical-sense are also trees in the data structure-sense :).

---

### Official Review · AnonReviewer1 · 2020-10-29
**no or little machine learning relevance, sparse result; meanwhile interesting research topic**

**Rating:** 3
**Confidence:** 4

**Review:**

This paper deals with the interesting question of how to simulate 3D shape of trees: given sets of 2D input images taken from drone cameras, the process consists of building point clouds, and textured triangle meshes, and creating a skinned cylindrical articulated rigid-body 3D shape. It is an important problem in artificial life, and has been investigated in the past, by e.g. L-system. In this work, a systematic pipeline is proposed and reasonable 3D shape result is demonstrated. I also enjoy the demo video and appreciate the amount of efforts devoted to annotate the tree data manually.
One the other hand, there are two main issues that prevent it from being considered in this conference venue. The first concerns the sparse result. With only one 3D shape being presented, it is difficult to understand the capacity of the proposed method; second, no or very weak machine learning and related technical contribution exist in the paper. Most of the text is devoted to 3D graphics and computer vision pipelines. Overall it seems unfit to ICLR audience.
There are a few other comments that I hope would be useful for the authors:
1. lack of technical contributions. Currently every module in the pipeline is off-the-shelf. The authors are encouraged to conconsider developing a new learning based method to tackle (maybe part of) the problem. The could dramatically change the feel of the current work.
2. more empirical studies. The current dataset seems to consist of only one single tree. It would be much more convincing to show multiple trees acquired at very different location/season/species, etc.

---

> ### Author Response · Authors · 2020-11-25
> **Response to Reviewer 1**
>
> We thank the reviewer for their comments and feedback.
>
> As the reviewer notes, techniques like L-systems have been used in the past to build procedural models of trees.
> It is our hope in this work to reconstruct a tree with real data, i.e. without proceduralism.
> Indeed, while the lack of additional empirical studies is a fair concern, our goal was to perform a study of how faithfully we could reconstruct that one real tree.
>
> While we have no new network architectures, we believe there is merit in the particular application of U-net with our various post-processing steps to aid in semi-automatic annotation of images.
> That said, it is true that various stages of the pipeline could be examined and improved; it is our hope that by making the pipeline and data available, various researchers may benefit from the framework within which they can do just this.

---

### Decision · Program_Chairs · 2021-01-07
**Final Decision**

**Decision:**

Reject

**Comment:**

Paper proposes and demonstrates a method to reconstruct 3d shape for a tree, from drone data.  While the reviewers all appreciated to work, all felt there were many shortcomings of the paper with respect to an ICLR audience:
(a)  no machine learning novelty
(b)  highly interactive data processing method
(c) only one example processed tree shown
(d) inadequate connections with relevant literature on 3d reconstruction, both general purpose, and examples applied to vegetation.
(e) incomplete presentation of the method:   no ablation studies, no listing of the times required for individual steps of the processing.

In view of these concerns, we have decided to reject the paper.  But we hope you find the reviewers' comments helpful, and make use of them in a revision of the work.